# TC-MoE: Augmenting Mixture of Experts with Ternary Expert Choice

**Shen Yan[1], Xingyan Bin[2], Sijun Zhang[2], Yisen Wang[3,4]∗, Zhouchen Lin[3,4,5]∗**
[1]Center for Data Science, Peking University
[2]Seed-Foundation-Model, ByteDance
[3]State Key Lab of General AI, School of Intelligence Science and Technology, Peking University
[4]Institute for Artificial Intelligence, Peking University
[5]Pazhou Laboratory (Huangpu), Guangzhou, Guangdong, China

## Abstract

The Mixture of Experts (MoE) architecture has emerged as a promising solution to reduce computational overhead by selectively activating subsets of model parameters. The effectiveness of MoE models depends primarily on their routing mechanisms, with the widely adopted Top-K routing scheme used for activating experts. However, the Top-K scheme has notable limitations, including unnecessary activations and underutilization of experts. In this work, rather than modifying the routing mechanism as done in previous studies, we propose the Ternary Choice MoE (TC-MoE), a novel approach that expands the expert space by applying the ternary set $\{-1, 0, 1\}$ to each expert. This expansion allows more efficient and effective expert activations without incurring significant computational costs. Additionally, given the unique characteristics of the expanded expert space, we introduce a new load balance loss and reward loss to ensure workload balance and achieve a flexible trade-off between effectiveness and efficiency. Extensive experiments demonstrate that TC-MoE achieves an average improvement of over 1.1% compared with traditional approaches, while reducing the average number of activated experts by up to 9%. These results confirm that TC-MoE effectively addresses the inefficiencies of conventional routing schemes, offering a more efficient and scalable solution for MoE-based large language models. Code and models are available at `https://github.com/stiger1000/TC-MoE`.

## 1 Introduction

In recent years, large language models (LLMs) (Brown et al., 2020; Touvron et al., 2023; Achiam et al., 2023) have demonstrated impressive performance across a wide range of domains. However, modern LLMs still face inefficiencies because they typically utilize all their parameters for every input token during both training and inference. This leads to substantially increased computational resource requirements as the models scale. To address these challenges, researchers have introduced the Mixture of Experts (MoE) architecture (Shazeer et al., 2017). The MoE architecture facilitates parameter scaling while maintaining reasonable computational costs. Unlike traditional dense models, MoE models incorporate a routing mechanism that selectively activates specific subsets of parameters for each input token. Recent advancements in MoE models (Jiang et al., 2024; Dai et al., 2024; Wu et al., 2024) have paved the way for scaling language models to unprecedented sizes while achieving remarkable performance improvements.

Within the MoE architecture, the routing mechanism plays a critical role since it significantly influences both the efficiency and effectiveness of model training. Traditional MoE frameworks, including GShard (Lepikhin et al., 2021), Switch Transformers (Fedus et al., 2022), and ST-MoE (Zoph et al., 2022), all employ the Top-K routing scheme. This method calculates the routing probability for each combination of experts and tokens. The K experts with the highest probabilities are activated for each token , with the final output being the weighted sum of their outputs.

---

∗Corresponding authors.

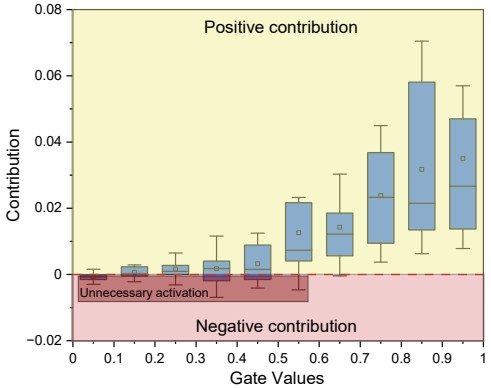 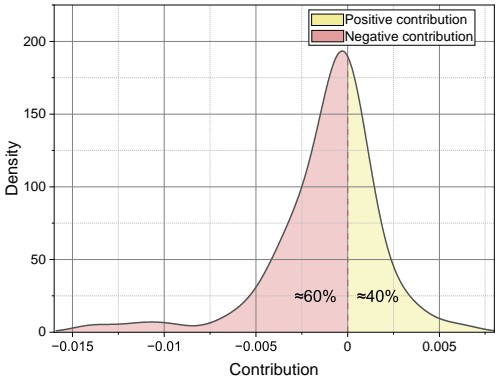

(a) Distribution of contributions from activated experts. The experts are categorized based on their gate values. This shows that some activations contribute negatively to the performance, indicating unnecessary activations.

(b) Distribution of contributions from experts with low gate values after flipping the sign of their outputs. The results demonstrate that some experts can positively impact performance when their output signs are flipped.

Figure 1: Analysis of the limitations in the conventional Top-K routing scheme in a model with 2 activated experts out of 8. We compute the contribution of each activated expert by measuring the difference in model performance when the activation is masked. More details are in Appendix A.

However, recent works (Zhou et al., 2022; Huang et al., 2024) combined with our experiments demonstrate the suboptimal nature of this routing scheme. We identify two key limitations:

- **Unnecessary Activations**: The Top-K scheme activates a fixed number of experts for each token, neglecting the possibility of adaptively choosing the number of activated experts. As shown in Figure 1a, some activated experts do not positively contribute to model performance, revealing redundant computations.

- **Underutilization of Experts**: The constraint of non-negative expert weights prevents full utilization of expert potential. Figure 1b demonstrates that approximately 40% of low-weight experts show positive contributions when their output signs are reversed.

In this work, we introduce the **Ternary Choice Mixture of Experts (TC-MoE)** to address these limitations. Unlike previous studies (Zhou et al., 2022; Yang et al., 2024; Huang et al., 2024) that focus on improving the routing scheme, we explore an innovative approach through expert space expansion. Inspired by ternary quantization techniques where weights are constrained to $\{-1, 0, 1\}$, we propose to create an expanded expert space through ternary multiplication of each original expert. As demonstrated in Figure 2b, assigning ternary choices to each expert generates expanded expert space with unchanged parameter counts of the experts. This expanded expert space empowers the router to learn sophisticated strategies during the training process, thereby alleviating the aforementioned limitations without modifying the routing scheme.

However, the expanded expert space exhibits unique characteristics differing from the original expert space. As illustrated in Figure 2b, parameter-sharing expert pairs coexist with zero-parameter experts that incur computational costs. These structural differences render traditional load balance loss ineffective. To address this, we propose a redesigned load balance loss for equitable workload distribution. Furthermore, we introduce a novel reward loss enabling flexible efficiency-effectiveness trade-offs through our analysis.

We conduct comprehensive evaluations of our method on multiple benchmarks. The results demonstrate that our TC-MoE outperforms existing approaches. Compared with the baseline, TC-MoE delivers an average performance gain of 1.1% alongside a 9% reduction in the average number of activated experts. Notably, under varying computational budgets, our method maintains consistent performance advantages over dynamic routing alternatives. These findings conclusively demonstrate the effectiveness of TC-MoE in overcoming the limitations of conventional routing schemes.

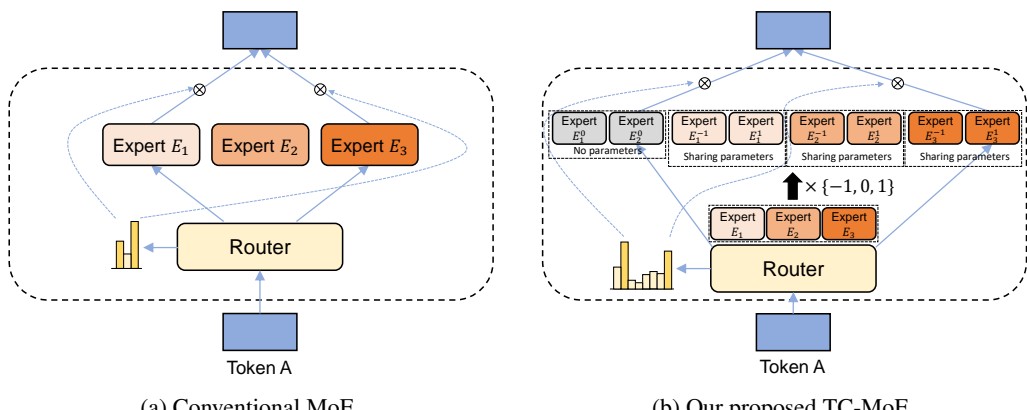

(a) Conventional MoE.  (b) Our proposed TC-MoE.

Figure 2: Comparison of the conventional MoE architecture and the proposed TC-MoE. The original expert space contains 3 experts, with the router activating 2 experts per token. By multiplying each expert with the ternary set $\{-1, 0, 1\}$, TC-MoE obtains an expanded expert space comprising 8 experts. Notably, 2 of these experts are parameter-free (we maintain 2 instead of 3 as 2 is sufficient for the router to activate any number from 0 to 2 of these experts). The remaining 6 experts are grouped into 3 parameter-sharing sets.

Our contributions can be summarized as follows:

1. We propose the TC-MoE, a novel method to overcome classical routing limitations in MoE architectures by expanding the expert space through multiplying the ternary set, achieving effective activation enhancement without routing scheme modification.

2. Given the unique load balance requirements in our expanded expert space, we redesign the load balance loss. Additionally, we introduce a novel reward loss to achieve a flexible trade-off between effectiveness and efficiency.

3. Our experimental results demonstrate that TC-MoE achieves superior performance with fewer activated parameters, confirming significant improvements in both effectiveness and efficiency over the baseline.

## 2 RELATED WORK

**Mixture of Experts Models.** MoE models (Jacobs et al., 1991; Jordan & Jacobs, 1994) have been extensively studied in artificial intelligence. The concept of using a trainable gating network to determine a sparse combination of experts is pioneered by the Sparsely-Gated MoE (Shazeer et al., 2017). Since then, numerous studies (Lepikhin et al., 2021; Fedus et al., 2022; Zoph et al., 2022; Jiang et al., 2024; Dai et al., 2024; Wei et al., 2024) have built upon this framework, demonstrating compelling empirical results by scaling MoE models to unprecedented sizes.

**Routing Schemes.** The MoE architecture relies on a routing module to determine the activation of experts, making the routing scheme a critical factor for MoE model performance. Early works (Shazeer et al., 2017; Fedus et al., 2022) employ the Top-K routing scheme, which calculates the routing probabilities for each expert and activates the Top-K experts with the highest probabilities. Recent studies have focused on improving routing schemes. Zhou et al. (2022) introduce the expert choice routing mechanism, which assigns equal capacity to every expert and allows tokens to compete for expert selection. Yang et al. (2024) propose a threshold-based router that uses a manually set threshold to control the number of activated experts for each token. Huang et al. (2024) also propose a threshold-based router but take it a step further by incorporating a dynamic loss to prevent activating too many experts.

**Heterogeneous Experts Design.** Unlike classical MoE frameworks that utilize feed-forward networks with the same configuration for all experts, recent works have explored the design of heterogeneous experts. Ainslie et al. (2023b) propose a heavy branch alongside a light branch, using a

router to select important tokens for processing through the heavy branch. Raposo et al. (2024) further refine this concept in the decoder-only setting by defining the light branch as a skip connection. Additionally, Zeng et al. (2024) introduce a set of null experts alongside ordinary experts, while Wang et al. (2024) explore more diverse strategies for integrating heterogeneous experts.

In this paper, we propose the TC-MoE, a novel method that complements existing research on routing mechanisms in MoE models. Our framework provides a comprehensive design for expert spaces by leveraging heterogeneous experts, thereby improving the overall performance and scalability of MoE architectures.

## 3 PROPOSED TC-MOE

In this section, we begin with an overview of the widely used Top-K routing mechanism in MoE models. We then introduce our Ternary Choice MoE, a method that expands the expert space with minimal computational overhead. Following this, we propose a new load balance loss to ensure effective load balance across the expanded expert space. Finally, we present a reward loss technique that achieves a flexible trade-off between efficiency and performance in our approach.

### 3.1 REVIEW OF TOP-K ROUTING MECHANISM

In a typical MoE architecture for transformer language models, Feed-Forward Network (FFN) layers are replaced with MoE layers. Each MoE layer consists of $N$ independent FFNs, referred to as experts, $\{E_1, E_2, \cdots, E_N\}$, along with a trainable router. Given a hidden representation $\mathbf{h} \in \mathbb{R}^d$ of the input token, the router computes the probability distribution over the experts as follows:

$$p(\mathbf{h}) = \mathrm{Softmax}(\mathbf{W}_g \cdot \mathbf{h} + \mathbf{b}_g), \tag{1}$$

where $\mathbf{W}_g \in \mathbb{R}^{N \times d}$ is a trainable weight matrix, and $\mathbf{b}_g \in \mathbb{R}^N$ is the bias term. Then the Top-K router selects the top $K$ experts with the highest probabilities for each input token. The gate values for the selected experts are set to the normalized probabilities, while those for the other experts are set to 0:

$$g_i(\mathbf{h}) = \begin{cases} p_i(\mathbf{h}) / \sum\limits_{j \in \mathcal{E}} p_j(\mathbf{h}), & i \in \mathcal{E} \\ 0, & i \notin \mathcal{E} \end{cases} \tag{2}$$

where $\mathcal{E}$ denotes the set of the top $K$ experts with the highest probabilities. The final output $O$ of the MoE layer is computed as the weighted sum of the outputs from the activated experts:

$$\mathbf{O} = \sum_{i \in \mathcal{E}} g_i(\mathbf{h}) \cdot E_i(\mathbf{h}). \tag{3}$$

### 3.2 TERNARY CHOICE

Although the Top-K routing scheme is widely used in MoE models, we identify two key limitations of this approach. As illustrated in Figure 1, the scheme exhibits unnecessary activations where some activated experts negatively affect model performance. It also fails to fully utilize existing experts, neglecting the potential benefits of contrasting expert outputs. While most previous studies have focused on routing scheme modifications, we propose the TC-MoE, which expands the expert space to provide richer activation options for the router. Specifically, as illustrated in Figure 2b, we expand the original expert space by applying the ternary set $\{-1, 0, 1\}$. This allows us to project each expert $E_i$ into three distinct experts $\{E_i^{-1}, E_i^0, E_i^1\}$, defined as follows:

$$E_i^1(\mathbf{h}) := E_i(\mathbf{h}), \quad E_i^0(\mathbf{h}) := 0, \quad E_i^{-1}(\mathbf{h}) := -E_i(\mathbf{h}), \quad \forall \mathbf{h} \in \mathbb{R}^d. \tag{4}$$

In our design, $E_i^1$ and $E_i^{-1}$ share parameters with $E_i$. While $E_i^0$ contains no parameters and remains identical across all experts. We further simplify by retaining only $E_1^0, \cdots, E_K^0$, as this is sufficient for the Top-K router to activate any number from $0$ to $K$ of these experts. As a result, TC-MoE has a total of $2N + K$ experts.

As $E_i^1$ and $E_i^{-1}$ share parameters with $E_i$, the only additional parameters and computational costs in our method come from the router component. With the number of experts increased to $2N +$

$K$, TC-MoE introduces $(N + K)d + N + K$ additional parameters and incurs $O((N + K)d)$ additional computational overhead. Importantly, these additions are negligible compared to the overall computational costs of the MoE block.

For simplicity, we define the sets of each type of expert as follows:

$$E^{-1} := \{E_i^{-1} | i \in [N]\}, \quad E^0 := \{E_i^0 | i \in [K]\}, \quad E^1 := \{E_i^1 | i \in [N]\} \tag{5}$$

Our method provides an alternative perspective for addressing the aforementioned limitations of the Top-K routing scheme. By incorporating $E^0$, the router can avoid unnecessary activations by activating experts from $E^0$, which do not contribute to the output and require no computation. Moreover, introducing $E^{-1}$ enables the router to explore the potential benefits of flipping the signs of expert outputs.

Furthermore, we find that making a small improvement to the Top-K routing scheme by always activating experts from $E^0$ is beneficial, which is described in detail in Appendix C.

### 3.3 LOAD BALANCE LOSS

In common MoE models (Fedus et al., 2022; Zoph et al., 2022), an auxiliary loss is typically introduced to encourage a balanced workload among experts. However, in our approach, experts are classified into two types: $E^1 \cup E^{-1}$, which incurs computational costs, and $E^0$, which does not incur any computational costs. Therefore, reasonable workload balance considerations in our scenario are as follows: (1) experts from $E^0$ do not need to be balanced with other experts since they do not contribute to computational costs, and (2) the sum of the workloads of expert $E_i^1$ and expert $E_i^{-1}$ should be balanced, as $E_i^1$ and $E_i^{-1}$ are distributed on the same device in scenarios involving expert parallelism (Lepikhin et al., 2021). Based on these considerations, we propose a new formulation for the load balance loss:

$$f_i = \frac{1}{KT} \sum_{j=1}^{T} \mathbb{1}\left(\text{Token } j \text{ selects expert } E_i^1 \text{ or } E_i^{-1}\right), \tag{6}$$

$$\overline{f} = \frac{1}{N} \sum_{i=1}^{N} f_i, \tag{7}$$

$$p_i = \frac{1}{T} \sum_{j=1}^{T} \left[ p_{E_i^1}(\mathbf{h}_j) + p_{E_i^{-1}}(\mathbf{h}_j) \right], \tag{8}$$

$$\mathcal{L}_{aux} = \sum_{i=1}^{N} \left( f_i - \overline{f} \right) \cdot p_i, \tag{9}$$

where $T$ is the sequence length, $f_i$ represents the sum of the activation frequencies of experts $E_i^1$ and $E_i^{-1}$, and $p_i$ denotes the sum of the average probabilities assigned to experts $E_i^1$ and $E_i^{-1}$.

### 3.4 FLEXIBLE TRADE-OFF BETWEEN EFFICIENCY AND EFFECTIVENESS

Since $E^0$ represents a special class of experts that incurs no computational costs, it is crucial to understand how the router learns to allocate gate values to these experts. Based on our analysis, we propose a novel auxiliary loss, termed the reward loss, to achieve a flexible trade-off between efficiency and effectiveness by tuning the activated ratio of experts from $E^0$.

During the backward pass, the gradient of the gate value for each expert is computed as follows:

$$\frac{\partial \mathcal{L}}{\partial g_i(\mathbf{h})} = \begin{cases} \left\langle \frac{\partial \mathcal{L}}{\partial \mathbf{O}}, E_i(\mathbf{h}) \right\rangle, & i \in \mathcal{E} \\ 0. & i \notin \mathcal{E} \end{cases} \tag{10}$$

For each activated expert $E_i$, the term $-\frac{\partial \mathcal{L}}{\partial g_i(\mathbf{h})}$ indicates the impact of increasing the gate value on reducing the loss function. Since the sum of the gate values is constrained to 1, a competitive

Table 1: Comparison of performance across evaluation benchmarks. "**Avg. K**" denotes the average number of activated experts that incurs computational costs during inference. "**#FLOPs ↓**" denotes the reduction ratio of FLOPs compared to the Top-K baseline. The **bold** number indicates the highest value for each benchmark.

| Pre-trained Dataset | Method | Avg. K | #FLOPs ↓ | ARC-Easy | BoolQ | MMLU | LAMBADA | HellaSwag | OpenBookQA | PIQA | SIQA | WinoGrande | Avg |
|---|---|---|---|---|---|---|---|---|---|---|---|---|---|
| | *Base model* | | | | | | | | | | | | |
| | Top-K | 2.00 | - | **57.03** | 58.75 | 25.24 | **50.40** | 42.76 | 39.40 | 68.17 | 43.91 | 52.72 | 48.71 |
| | Random drop | 1.85 | 5.4% | 56.48 | 58.62 | 25.35 | 50.13 | 42.83 | 39.00 | **69.53** | 44.68 | 51.30 | 48.66 |
| | Top-P | 1.99 | 0.3% | 55.26 | **59.54** | **25.74** | 50.30 | 42.22 | 41.00 | 68.66 | 43.55 | 53.20 | 48.83 |
| RedPajama | TC-MoE | 1.82 | 6.5% | **57.03** | 59.20 | 25.58 | 50.16 | **43.51** | **42.00** | 68.66 | **44.88** | **54.85** | **49.54** |
| | *Fine-grained base model* | | | | | | | | | | | | |
| | Top-K | 4.00 | - | 56.69 | 55.35 | 25.16 | 50.16 | 42.72 | 39.6 | **68.93** | 44.11 | **52.49** | 48.36 |
| | TC-MoE | 3.87 | 2.3% | **57.58** | **58.56** | **26.80** | **50.46** | **43.16** | **41.80** | 68.28 | **45.19** | 52.09 | **49.32** |
| | *Tiny model* | | | | | | | | | | | | |
| | Top-K | 2.00 | - | 55.13 | 56.76 | 26.02 | 48.32 | 46.46 | 37.60 | 71.33 | 44.68 | 52.41 | 48.75 |
| FineWeb | TC-MoE | 1.83 | 5.8% | **55.93** | **58.53** | **26.2** | **48.85** | **46.65** | **41.20** | **71.71** | **46.32** | **53.99** | **49.93** |
| | *Base model* | | | | | | | | | | | | |
| | Top-K | 2.00 | - | 60.19 | 50.76 | 26.46 | 53.95 | 53.23 | 43.00 | **74.48** | 45.60 | 55.33 | 51.44 |
| | TC-MoE | 1.86 | 5.1% | **60.56** | **57.4** | **26.67** | **54.01** | **54.05** | **44.00** | 73.45 | **47.24** | **56.12** | **52.61** |

dynamic arises among the activated experts. Experts that significantly contribute to reducing the loss function are assigned higher gate values, as verified in Figure 1a.

Following Equation 10, for activated expert $E_i^0$, we have

$$\frac{\partial \mathcal{L}}{\partial g_{E_i^0}(\mathbf{h})} = \left\langle \frac{\partial \mathcal{L}}{\partial \mathbf{O}}, E_i^0(\mathbf{h}) \right\rangle = \left\langle \frac{\partial \mathcal{L}}{\partial \mathbf{O}}, \mathbf{0} \right\rangle = 0. \tag{11}$$

This indicates that expert $E_i^0$ has no impact on reducing the loss function. Therefore, when competing with other activated experts, expert $E_i^0$ tends to receive higher gate values than experts with negative impacts but lower gate values than those with positive impacts. This effectively helps avoid unnecessary activations.

Based on the above analysis, we propose extending our method to achieve a flexible trade-off between efficiency and effectiveness. Specifically, we manually assign a negative value to $\frac{\partial \mathcal{L}}{\partial g_{E_i^0}(\mathbf{h})}$ instead of 0, thereby giving expert $E_i^0$ a positive contribution in reducing the loss function. Consequently, the router will learn to promote the activation of these experts, while selectively deactivating other types of experts with minimal positive contributions. To achieve this, we introduce a new auxiliary loss, termed the reward loss, defined as follows:

$$\mathcal{L}_{rwd} = -\frac{1}{T} \sum_{i=1}^{K} \sum_{j=1}^{T} g_{E_i^0}(\mathbf{h}_j), \tag{12}$$

where $T$ is the sequence length, and $g_{E_i^0}(\mathbf{h}_j)$ represents the gate values of expert $E_i^0$ on token $\mathbf{h}_j$.

Linearly combining the language modeling loss ($\mathcal{L}_{lm}$), the load balance loss, and the reward loss, we yield the total loss, formulated as follows:

$$\mathcal{L} = \mathcal{L}_{lm} + \alpha_1 \mathcal{L}_{aux} + \alpha_2 \mathcal{L}_{rwd}, \tag{13}$$

where $\alpha_1$ is a hyper-parameter known as the load balance factor, and $\alpha_2$ is a hyper-parameter known as the reward factor.

# 4 EXPERIMENTS

## 4.1 EXPERIMENTAL SETTINGS

**Pre-trained Datasets.** We train our models using the RedPajama dataset (Computer, 2023) and the FineWeb dataset (Penedo et al., 2024). The RedPajama dataset includes diverse sources such as Common Crawl (CC), C4, Wikipedia, Github, books, arxiv, and Stackexchange. The FineWeb dataset is an open-source, high-quality training dataset consisting of cleaned and deduplicated English web data from CC. In our experiments, all models are trained on 100B tokens.

**Architecture.** We employ a decoder-only transformer model, primarily based on the LLaMA architecture (Touvron et al., 2023). Each transformer layer includes both an attention layer and an

Table 2: Configurations of our MoE models.

| Model | #Layers | #Hidden Size | #Heads | #KV Heads | #Intermediate Size | #Activated Experts/ #Total Experts | #Activated Params/ #Total Params |
|---|---|---|---|---|---|---|---|
| Tiny | 24 | 768 | 12 | 2 | 2048 | 2/8 | 298M/978M |
| Base | 32 | 1024 | 16 | 2 | 2816 | 2/8 | 681M/2.3B |
| Fine-grained base | 32 | 1024 | 16 | 2 | 1280 | 4/16 | 631M/2.1B |

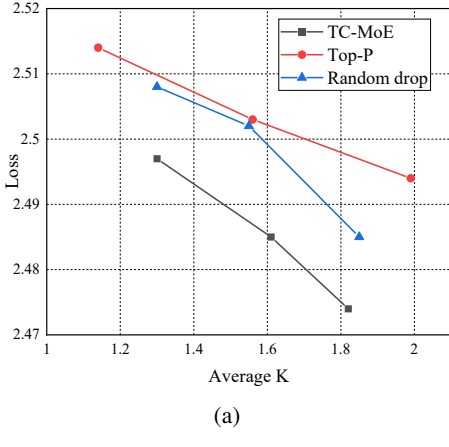
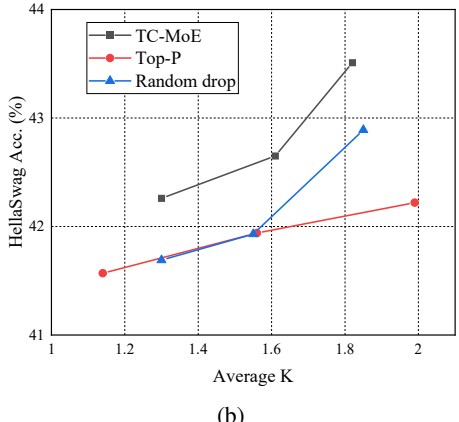

(a)                                                                        (b)

Figure 3: Comparison of (a) the language modeling loss and (b) the accuracy on HellaSwag under different budgets for the average number of activated experts. The results demonstrate TC-MoE outperforms other competitors under all settings.

**MoE layer.** RMSNorm (Zhang & Sennrich, 2019) is applied to the inputs of both attention layers and MoE layers. Within the attention layer, we adopt the Group-Query Attention (GQA) (Ainslie et al., 2023a). Additionally, RMSNorm is used to normalize each key vector (Dehghani et al., 2023). Each FFN expert employs the SwiGLU activation function (Shazeer, 2020). In our experiments, We employ three types of models: tiny, base, and fine-grained base, as summarized in Table 2. We use the same tokenizer as GPT-NeoX-20B (Black et al., 2022), which has a vocabulary size of 50257.

**Competitors.** We pre-train three baseline methods alongside our proposed TC-MoE:

1. **Top-K**: A standard Top-K routing scheme that activates the top $K$ experts for each token. We select $K = 2$ or $K = 4$ as these are the most common configurations in modern MoE architectures (Zoph et al., 2022; Jiang et al., 2024; Wei et al., 2024; Wu et al., 2024).

2. **Random drop**: A variant of the Top-K routing scheme that, with probability $p$, does not activate the expert with the second highest probability.

3. **Top-P**: The Top-P routing scheme (Huang et al., 2024), which activates the smallest set of experts whose cumulative probabilities surpass a threshold $P$ for each token.

4. **TC-MoE (ours)**: It expands the expert space and adopts the standard Top-K routing scheme to activate the top $K$ experts within this expanded expert space for each token.

The Top-K baseline adopts a fixed number of activated experts, whereas Random drop, Top-P, and TC-MoE allow for a flexible trade-off between effectiveness and efficiency by tuning specific hyper-parameters. Details of these hyper-parameters are provided in Appendix B.

**Evaluation.** We evaluate these models on seven benchmarks: ARC (Clark et al., 2018), BoolQ (Clark et al., 2019), MMLU (Hendrycks et al., 2021), LAMBADA (Paperno et al., 2016), HellaSwag (Zellers et al., 2019), OpenBookQA (Mihaylov et al., 2018), PIQA (Bisk et al., 2020), SIQA (Sap et al., 2019), and WinoGrande (Sakaguchi et al., 2021). These tasks assess the model performance on logical reasoning, language understanding, commonsense reasoning, and knowledge utilization. Additionally, we measure the average number of activated experts that incur computational costs to demonstrate the efficiency of each model. Note that in our TC-MoE, only the activations of $E^{-1}$ and $E^1$ are counted since $E^0$ incurs no computational costs. For simplicity, we refer to the average

Table 3: Ablation study on the contribution of different types of experts. "Multiplication Set" denotes the set used to multiply the original expert space, "Average K" denotes the average number of activated experts. Specifically, {1} represents the Top-K baseline.

| Multiplication Set | Average K | #FLOPs ↓ | Average Performance |
|---|---|---|---|
| {1} | 2.00 | - | 48.71 |
| {−1, 1} | 2.00 | 0.0% | 49.00 (+0.29) |
| {0, 1} | 1.81 | 6.9% | 49.23 (+0.52) |
| {−1, 0, 1} | 1.82 | 6.4% | 49.54 (+0.83) |

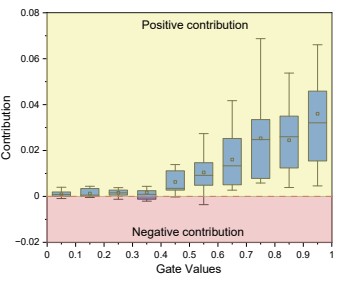

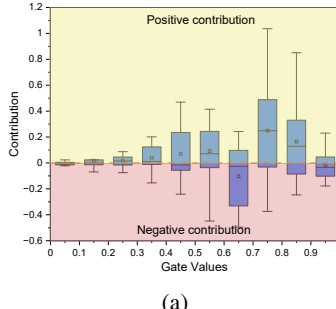

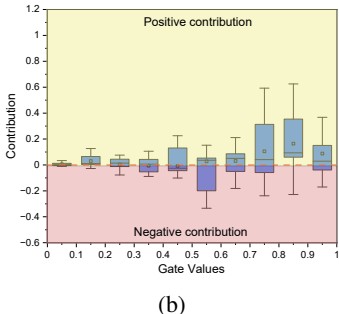

(a)           (b)

Figure 4: Distribution of contributions from activated experts in TC-MoE on pre-trained data.

Figure 5: Distribution of contributions from activated experts in (a) baseline and (b) TC-MoE on ARC-Easy. The results show a significant alleviation of unnecessary activations by TC-MoE.

number of activated experts as the average number of activated experts that incur computational costs in the following sections.

## 4.2 MAIN RESULTS

Table 1 summarizes the performance of various models across different evaluation benchmarks. The results highlight the superior performance of the proposed TC-MoE.

Specifically, when the base model is pre-trained on the RedPajama dataset, TC-MoE outperforms competitors on ARC-Easy, HellaSwag, OpenBookQA, SIQA, and WinoGrande, while achieving comparable results on BoolQ, MMLU, LAMBADA, and PIQA. Notably, TC-MoE achieves an average accuracy of 49.54%, surpassing the Top-K baseline by 0.83%, Random drop by 0.88% and Top-P by 0.71%. For the fine-grained base model pre-trained on the RedPajama dataset, TC-MoE also outperforms the Top-K baseline, improving the average accuracy by 0.96%.

When pre-training on the FineWeb dataset, TC-MoE demonstrates even greater accuracy improvements. For the tiny model, TC-MoE surpasses the Top-K baseline by 1.18%. Similarly, for the base model, TC-MoE outperforms the Top-K baseline by 1.17%.

Beyond improved accuracy, TC-MoE consistently demonstrates greater efficiency. Specifically, it reduces the average number of activated experts by 9.0% and the required FLOPs by 6.5% compared to the Top-K baseline when using the base model pre-trained on the RedPajama dataset. On the FineWeb dataset, TC-MoE reduces the average number of activated experts by 7.0% and the required FLOPs by 5.1%. These results demonstrate that our method achieves significant gains in both effectiveness and efficiency over the Top-K baseline.

Additionally, we conduct a thorough comparison of these methods under different budgets for the average number of activated experts. The results are shown in Figure 3. The figure demonstrates that TC-MoE consistently outperforms the other two competitors. Notably, in terms of the language modeling loss, TC-MoE reduces the loss by approximately 0.017 compared to competitors. In terms of the accuracy on HellaSwag, TC-MoE improves by up to 0.7% compared to other methods.

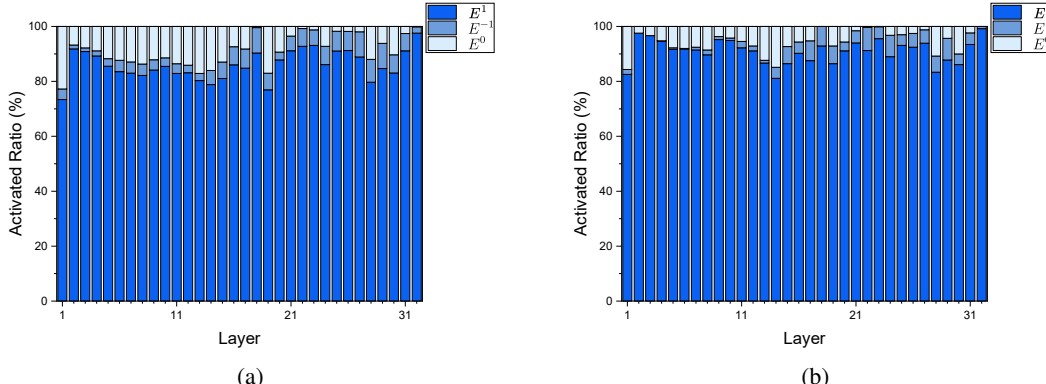

Figure 6: The activated ratios of different types of experts across layers on (a) the pre-trained data and (b) the test data (ARC-Easy). The results show a similar activated pattern on different data.

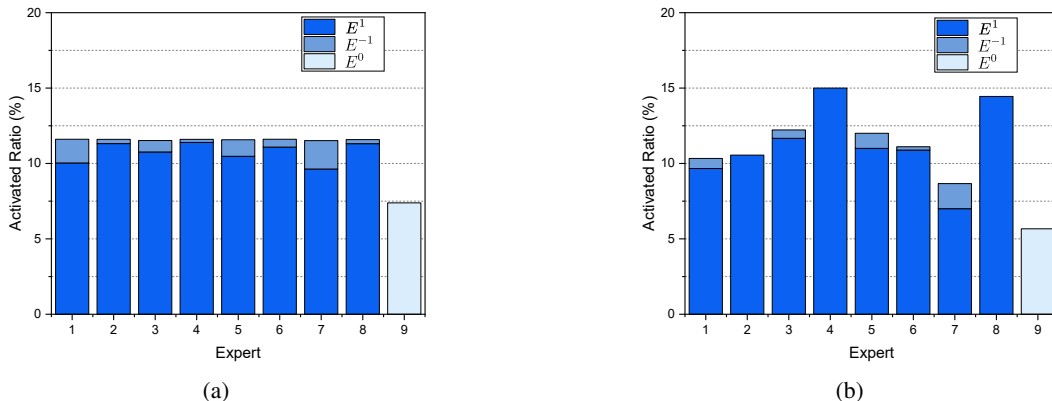

Figure 7: The activated ratios of different experts in layer 16 on (a) the pre-trained data and (b) the test data (ARC-Easy). The results show the effectiveness of our load balance loss during both training and inference.

### 4.3 ABLATION STUDY

We conduct an ablation study by evaluating the performance of our method using only a subset of $\{-1, 0, 1\}$ to multiply the original expert space. As demonstrated in Table 3, expanding the expert space with either $\{-1, 1\}$ or $\{0, 1\}$ improves the model performance. Specifically, expanding the expert space with $\{-1, 1\}$ results in a 0.29% average performance increase, while maintaining the average number of activated experts at 2.0. When expanding the expert space with $\{0, 1\}$, the average performance increases by 0.52%, and the average number of activated experts is reduced by 0.19. When expanding the expert space with the complete set $\{-1, 0, 1\}$, the model achieves the best performance, with both improved results and a reduced number of activated experts. In summary, both the $E^{-1}$ and $E^0$ types contribute to the improvement in model performance, while the $E^0$ type also significantly enhances model efficiency.

### 4.4 ANALYSIS

**Unnecessary Activations.** We investigate the effect of our method on reducing unnecessary activations. Figure 4 shows the distribution of contributions from activated experts in the TC-MoE model on the pre-trained data. Compared to the distribution of contributions in the baseline model, shown in Figure 1a, our TC-MoE significantly reduces the occurrence of unnecessary activations. Additionally, when analyzing unnecessary activations on ARC-Easy, we observe a significantly large number of activations that contribute negatively. As shown in Figure 5, TC-MoE demonstrates an even greater reduction of unnecessary activations in this scenario.

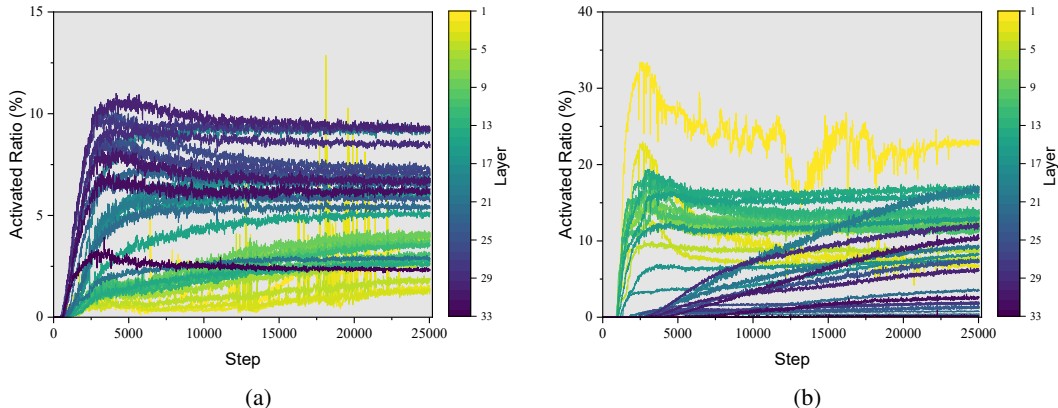

Figure 8: The changing curves of activated ratios of (a) type $E^{-1}$ and (b) type $E^0$ of different layers.

**Activated Ratio of Different Types of Experts.** To analyze the activated ratio of different types of experts, we visualize the activated ratios across layers in the TC-MoE model. The results are shown in Figure 6. We observe that the $E^1$ type has the highest activated ratio, indicating its major contribution to the output. Additionally, the model spontaneously learns to allocate an activated ratio of totally around 20% to the $E^0$ type and the $E^{-1}$ type, highlighting their necessity for more powerful routing. The activation ratios on ARC-Easy are similar to those on the pre-trained data, demonstrating the generalization of our method.

The distribution of activated ratios varies significantly across different layers. Figure 8 shows the changing curves of activated ratios across layers throughout training. We find that the $E^{-1}$ and $E^0$ types exhibit contrasting activation patterns across layers. Specifically, the model activates a higher proportion of $E^{-1}$ experts in deeper layers, while $E^0$ experts are predominantly activated in shallow layers.

**Load Balance.** To explore the effectiveness of our load balance loss, we visualize the activated ratios of different experts in TC-MoE during training. Figure 7 shows the activated ratios in layer 16. To observe the actual load balancing distribution, we stack the activated ratios of each $E_i^{-1}$ and $E_i^1$ pair, as these experts are distributed on the same device when involving expert parallelism. Additionally, we plot the sum of the activated ratios of experts from $E^0$ at the position of expert 9, since these experts do not contribute to computational costs. The results demonstrate that TC-MoE achieves near-perfect workload balance on the pre-trained data. The sum of the workloads of experts $E_i^1$ and experts $E_i^{-1}$ are balanced, each around 11.5%. Additionally, the activated ratios of expert $E_i^1$ and expert $E_i^{-1}$ are not fixed but are instead learned dynamically by the model. Furthermore, experts from $E^0$ do not participate in the load balancing, allowing the model to activate $E^0$ without any constraints. On ARC-Easy, we observe a slight deviation with maximum workload at 15.0% versus minimum at 8.5%.

## 5  CONCLUSIONS

In this paper, we present the Ternary Choice MoE (TC-MoE), a novel approach that addresses the limitations of traditional Top-K routing in MoE architectures. By expanding the expert space through multiplying each expert with the ternary set $\{-1, 0, 1\}$, we introduce greater flexibility and diversity into expert activations without significant computational overhead. Our approach enhances expert utilization, effectively mitigating both unnecessary activations and expert underutilization in conventional MoE models. Extensive experiments demonstrate that TC-MoE achieves consistent improvements in effectiveness and efficiency, surpassing existing methods across multiple benchmarks. These results highlight the potential of TC-MoE as a scalable, computationally efficient method for MoE models. We believe this work provides new perspectives for further development in the design and optimization of MoE models, paving the way for more advanced and resource-efficient large-scale models.

ACKNOWLEDGEMENTS

Zhouchen Lin and Yisen Wang were supported by National Key R&D Program of China (2022ZD0160300). Zhouchen Lin was also supported by the NSF China (No. 62276004). Yisen Wang was also supported by National Natural Science Foundation of China (92370129, 62376010), and Beijing Nova Program (20230484344, 20240484642).

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

## A    DETAILS OF CALCULATING THE CONTRIBUTION OF EACH ACTIVATION

To evaluate the effectiveness of the routing scheme, we conduct experiments to analyze the impact of routing decisions on model outputs. It is important to note that we are not discussing the contribution of each expert, but rather the effect of each activation decision made by the router.

Specifically, we first randomly select 15 sequences from the training set of RedPajama (Computer, 2023) and the test set of ARC-Easy (Clark et al., 2018), respectively. The average sequence length of samples in the training set is 1608, while it is 30 in the test set, We use the language modeling loss on these sequences to measure the quality of model outputs. For a specific activation $A$, a straightforward method to obtain its contribution is to forward the model on the input sequence twice: once with this activation and once masking this activation. We then calculate the difference in the value of the loss function. This can be formulated as follows:

$$\text{Contribution}_A := \mathcal{L}(M_{\backslash\{A\}}(x)) - \mathcal{L}(M(x)), \tag{14}$$

where $x$ denotes the input sequence, $\mathcal{L}$ denotes the loss function, $M$ denotes the function of the model, and $M_{\backslash\{A\}}$ denotes the function of the model when masking the activation $A$. For sequences sampled from the pre-trained data, we compute the loss across all positions, whereas for sequences sampled from the test data, we calculate the loss only over the tokens constructing the answer. When masking activation $A$ results in a higher loss, we calculate a positive contribution for activation $A$. This indicates that the activation decision has a beneficial impact on model performance.

In practice, we observe empirically that the impact of masking a single activation is too small to analyze due to numerical errors. Therefore, we alternatively categorize the activations into groups based on their gate values and calculate the contribution of each group of activations. Specifically, we divide the gate values from 0 to 1 into 10 intervals: 0 to 0.1, 0.1 to 0.2, ..., 0.9 to 1.0. Nevertheless, the unequal size of different groups still makes it unfair to compare the contributions across groups. To address this, we randomly select the same number of activations to mask within each group for a fair comparison. Specifically, we mask 20 activations in each layer of each group for sequences sampled from the training set, and 5 activations in each layer of each group for sequences sampled from the test set. We then calculate the loss difference as shown in Equation 14.

For the experiments involving the flipping of expert output signs, we use a similar method. For a specific activation $A$, we forward the model on the input sequence twice: once with this activation and once with the flipped sign activation, then calculate the difference in the value of the loss function. We define this as:

$$\text{Contribution}_{-A} := \mathcal{L}(M_{\backslash\{A\}, \cup\{-A\}}(x)) - \mathcal{L}(M(x)), \tag{15}$$

where $M_{\backslash\{A\}, \cup\{-A\}}$ denotes the function of the model when the sign of activation $A$ is flipped. We randomly flip the sign of 20 activations with gate values lower than 0.2 in each layer, obtaining the distribution shown in Figure 1b.

## B    HYPER-PARAMETERS

We use the AdamW optimizer with exponential decay rates $\beta_1 = 0.9$ and $\beta_2 = 0.95$, and apply a weight decay of 0.1 throughout training. The learning rate is warmed up linearly from 0 to 3e-4 during the initial 10% of training, then decays to 3e-5 following a cosine decay schedule for the remaining steps. We set the sequence length to 2048 and the global batch size to 2048. Batch size warmup is synchronized with the learning rate warmup schedule.

To achieve a flexible trade-off between effectiveness and efficiency for Random drop, Top-P, and TC-MoE, we set hyperparameters as follows:

- **Random drop**: We set the drop probability $p$ to 15%, 45%, and 70% to achieve average activation numbers of 1.85, 1.55, and 1.30, respectively.
- **Top-P**: We set the threshold $P$ to 0.4 as in the original paper (Huang et al., 2024), and the dynamic loss weight to 1e-5, 2e-5, and 5e-5 to achieve average activation numbers of 1.99, 1.56, and 1.14, respectively.
- **TC-MoE**: We set the load balance factor $\alpha_1$ to 0.01, and the reward factor $\alpha_2$ to 0, 1e-5, and 2e-5 to achieve average activation numbers of 1.82, 1.61, and 1.30, respectively.

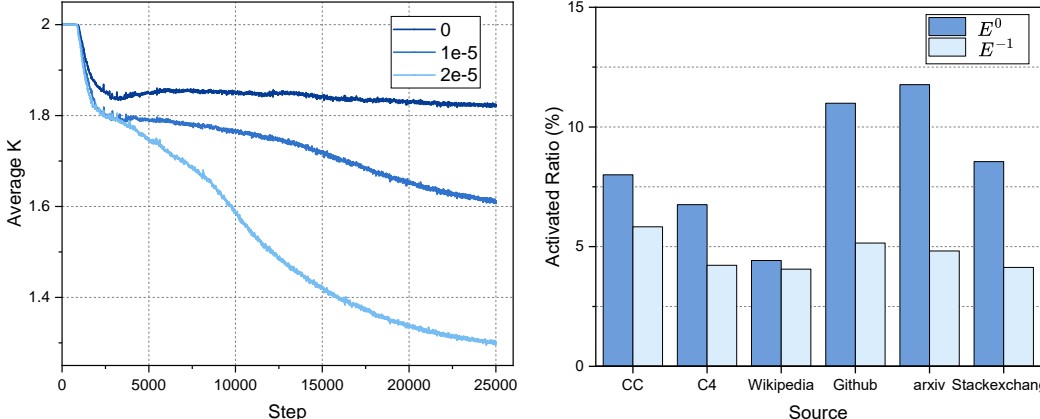

Figure 9: The changing curves of the average number of activated experts when varying the reward factor.

Figure 10: The activated ratios across different sources of the training data.

For initialization, we adopt an initializer range of 0.006. The weight matrix $W_g$ of the router is also initialized with a standard deviation of 0.006. The bias term $b_g \in \mathbb{R}^{2N+K}$ of the router is set such that experts of type $E^1$ have a bias of 0, those of type $E^{-1}$ have a bias of $-1$, and those of type $E^0$ have a bias of $-10$. This particular initialization strategy is designed to guide the router to focus primarily on type $E^1$ during the early stages of the training process.

## C  IMPROVING THE TOP-K ROUTING SCHEME

Intuitively, experts from $E^0$ share similarities with attention sinks (Xiao et al., 2024). The intuition behind attention sinks lies in the fact that the attention scores calculated by the Softmax operation sum up to 1 for all tokens. As a result, the model naturally learns to assign useless attention scores to sink tokens. Similarly, in MoE models, the router uses the Softmax operation to calculate gate values. In this context, we believe that experts from $E^0$ serve as sink experts to collect unneeded gate values.

However, within the conventional Top-K routing mechanism, as illustrated in Equation 10, only the activated experts participate in the competition for gate values. This implies that experts from $E^0$ can collect useless gate values only when they are among the Top-K experts. To address this, we refine our design by consistently activating experts of type $E^0$, enabling them to participate fully in the competition for gate values and more effectively serve as sinks. Specifically, we revise the activation set $\mathcal{E}$ by taking its union with $E^0$. The updated calculation of gate values is formulated as follows:

$$g_i(\mathbf{h}) = \begin{cases} p_i(\mathbf{h}) / \sum_{j \in \mathcal{E} \cup E^0} p_j(\mathbf{h}), & i \in \mathcal{E} \cup E^0 \\ 0, & i \notin \mathcal{E} \cup E^0 \end{cases} \tag{16}$$

where $\mathcal{E}$ denotes the set of the top $K$ experts with the highest probabilities.

## D  EFFECT OF THE REWARD LOSS

We also investigate the effect of our designed reward loss. As illustrated in Figure 9, we vary the reward factor from 0 to 2e-5, The average number of activated experts shows different changing curves. By increasing the reward factor, we encourage the model to select experts from $E^0$, which incur no computational costs. Consequently, the model tends to have a lower average number of activated experts. Specifically, the average number of activated experts converges to 1.82 when the reward factor is 0 and to 1.30 when the reward factor is 2e-5. These results demonstrate that tuning the reward factor enables a flexible trade-off between effectiveness and efficiency.

# E    ACTIVATED RATIO ON DIFFERENT SOURCES

We also investigate the activated ratio on different sources of the training data. The results are shown in Figure 10. We observe that the activated ratio of $E^0$ exhibits significant variance across different sources. Notably, The activated ratio of $E^0$ is around 11% on data from Github and arxiv, while it is only 4% on data from Wikipedia. Similarly, the activated ratio of $E^{-1}$ also varies across different sources. The activated ratio of $E^{-1}$ is only 4% on data from Wikipedia and Stackexchange, while it is 6% on data from CC. The variance across different sources indicates the specialization of experts from $E^0$ and $E^{-1}$.

