# OpenReview forum: "TC-MoE: Augmenting Mixture of Experts with Ternary Expert Choice"
_ICLR.cc/2025/Conference — ICLR 2025 Poster_

### Official Review · Reviewer_6GcM · 2024-10-26

**Soundness:** 3
**Presentation:** 3
**Contribution:** 3
**Rating:** 6
**Confidence:** 5

**Summary:**

This paper proposes a scheme for improving MoE models by using the idea of ternary quantization over the experts. The core idea of this paper is based on their observation that the Top-k scheme underutilizes experts and does not have a way of dynamically selecting the number of experts for a particular token. Their scheme proposes to expand the number of experts available by converting the existing experts three types of experts : null, same, negative. In order to deal with the new challenges this scheme brings about, they design a new load balancing and reward loss. The load balancing loss simply extends the standard loss by accounting for the fact that the expert and the negation of that expert are really the same expert and thus on the same device and so their total workload needs to be balanced. Through experiments, they observe this scheme outperforms standard MoE models by about 1% in flop matched settings on downstream tasks.

**Strengths:**

In my opinion, the main strength of this paper is the simplicity of the proposed MoE augmentation scheme. Their observation regarding
the negative contribution of some experts is interesting though it is contrary to the idea that dense training / inference is better [1].
Their baseline comparisons are correct although I believe comparing simply with Top-k = 2 would be enough and the compute could have been allocated better elsewhere i.e training larger models. While there is limited explanation for why the ternary quantization works, they have good empirical observations. If the schemes load balancing is better than the top-k load balancing loss, then this approach is indeed worth scaling up in my opinion.

[1] - Pan, Bowen, et al. "Dense Training, Sparse Inference: Rethinking Training of Mixture-of-Experts Language Models." arXiv preprint arXiv:2404.05567 (2024).

**Weaknesses:**

The performance improvement is quite marginal and since the evaluation is done a relatively small models, we do not know if this technique will scale to larger models. I also did not find any theoretical / intuitive justification for why null experts or negative experts should help. In the questions subsection, I have included one potential reason but it would be interesting to know the intuition the authors had for developing this scheme.
The evaluation is done on limited benchmarks and the impact on realistically sized models is unknown.  A model with 2.3B total parameters has approximately only 700M activated parameters with a Top-k=2 routing scheme. Did the results hold on any smaller models trained or models trained on a more higher quality pretraining dataset like FineWeb? Can the authors evaluate on MMLU as well?. I would have also liked to see the reduction in FLOPS compared to the standard baseline and how their scheme is impacted by fine-grained experts. My intuition is that it would be less effective since their is more capacity within the base MoE to learn different types of experts and thus the need for negated or null experts might be less.

Typos and grammar errors
Line 105 - of classical routing scheme

**Questions:**

Why did the authors not release their code? How can I verify their results in this paper?
In Fig 2B I can see that there is no null expert for Expert 3. This is different from what the text says that the total expert count = 3 x base_num experts. Is this just an illustration error?

Is this paper suggesting that a dense forward pass is worse than a sparse forward pass? Personally, I have performed several experiments with varying the number of K over standard dense MoE models during training and always found performance to monotonically increase with the number of activated experts. It would be beneficial to be able to add some experiments where the authors can show activation how the performance on a benchmark task(s) is impacted by increasing top-k by 1 over some useful range.

How are the results in Fig 1a computed? Are these results on benchmark scores evaluated for a particular setting of K? A more detailed caption would be helpful for future readers.

How is the computation for the total number of experts done? It says on Line 210 it is $2N + K$ but if each expert is expanded to three experts, isn't that simply $3N$? Please clarify this.


Do the authors find any connection with null experts to attention sinks [1]? It would be interesting to have a discussion of that as I believe something similar to the idea of attention sinks is happening where instead of randomly routing to a useless expert (in the case of fixed top-k) the model can dynamically choose the number of experts needed for each token.

[1] - Xiao, G., et al. "Efficient Streaming Language Models with Attention Sinks, 2023." URL http://arxiv. org/abs/2309 17453.

---

> ### Author Response · Authors · 2024-11-21
> **Response to Reviewer 6GcM (1/3)**
>
> * **Weaknesses**:
>
> 1. > The performance improvement is quite marginal and since the evaluation is done a relatively small models, we do not know if this technique will scale to larger models.
>
>     We understand the importance of scaling to larger models.
>     Due to computational resource constraints,
>     we focus our experiments on the current scale.
>     Nevertheless, we strongly believe that the underlying intuition of our approach remains applicable and beneficial for larger models as well.
>
>     To better evaluate our method,
>     we have included additional experiments on models with a broader range of settings.
>     These experiments, as shown in Tab. 1 of the revised paper, demonstrate much more significant performance improvements.
>
> 2. > I also did not find any theoretical / intuitive justification for why null experts or negative experts should help. In the questions subsection, I have included one potential reason but it would be interesting to know the intuition the authors had for developing this scheme.
>
>     Thanks for your insightful comments.
>     Below we discuss the intuitive justifications for why null experts or negative experts are beneficial.
>
>     * Null experts
>
>         We agree that null experts share similarities with attention sinks.
>         The intuition behind attention sinks is that the attention scores calculated by the Softmax operation sum up to one for all tokens.
>         As a result, the model naturally learns to assign useless attention scores to sink tokens.
>         Similarly, in MoE models, the Softmax operation is used by the router to calculate gate values.
>         Therefore, we believe that null experts serve as sink experts in this case to collect unneeded gate values.
>
>         This concept plays a key role in the design of our method.
>         In Appendix A.3,
>         we propose an improvement to TC-MoE that enhances the behavior of null experts as sink experts.
>         In standard MoE models, only the Top-K experts receive non-zero gate values (see Equation 2).
>         This means that null experts are effective only when they are among the Top-K experts.
>         To address this, we modify the design by always activating null experts, allowing them to participate fully in the competition for gate values and better serve as sinks.
>
>         Moreover, null experts also contribute by acting as "skip connections" in the MoE module.
>         Our observation is that some experts, when activated,
>         contribute negatively to the model output.
>         By learning to activate null experts instead of these potentially harmful ones,
>         the model can improve both effectiveness and efficiency.
>
>     * Negative experts
>
>         The intuition behind negative experts comes from the observation that specialized experts may exhibit conflicting behaviors.
>         As supported by recent studies [1] and our own findings,
>         some experts may be harmful when processing specific tokens,
>         and directly contrasting their outputs can yield positive impacts.
>         By incorporating negative experts, the router gains the ability to strategically leverage these contrasting outputs,
>         ultimately enhancing the overall performance and robustness of the model.
>
>     While our current work does not provide a theoretical guarantee for the effectiveness of this approach,
>     our experimental results validate its practical benefits.
>     We aim to develop a rigorous theoretical framework to support these intuitions in future research.
>
>
> 3. > Did the results hold on any smaller models trained or models trained on a more higher quality pretraining dataset like FineWeb?
>
>     To address your concern,
>     we conduct additional experiments on smaller models and models pre-trained on FineWeb.
>     We show the results below:
>
>     | Method | Avg. K | \#FLOPs $\downarrow$  | ARC-Easy | BoolQ | MMLU | LAMBADA | HellaSwag | OpenBookQA | PIQA | SIQA | WinoGrande | Avg
>     |:-------|:-|:-|:-|:-|:-|:-|:-|:-|:-|:-|:-|:-|
>     | Top-K (Tiny) | 2.00 | - | 55.13 | 56.76 | 26.02 | 48.32 | 46.46 | 37.60 | 71.33 | 44.68 | 52.41 | 48.75
>     | TC-MoE (Tiny) | 1.83 | 5.8\% | 55.93 | 58.53 | 26.2 | 48.85| 46.65 | 41.20 | 71.71 | 46.32 | 53.99 | 49.93
>     | Top-K (Base) | 2.00 | - | 60.19 | 50.76 | 26.46 | 53.95 | 53.23 | 43.00 | 74.48 | 45.60 | 55.33 | 51.44
>     | TC-MoE (Base) | 1.86 | 5.1\% | 60.56 | 57.4 | 26.67 | 54.01 | 54.05 | 44.00 | 73.45 | 47.24 | 56.12 | 52.61
>
>     These results confirm that our improvements hold across different model scales and pretraining datasets.
>     We have detailed these additional experiment settings and results in the revised paper.
>
> References:
> [1] - Shi, Chufan, et al. "Unchosen Experts Can Contribute Too: Unleashing MoE Models' Power by Self-Contrast." arXiv preprint arXiv:2405.14507 (2024).

---

> ### Author Response · Authors · 2024-11-21
> **Response to Reviewer 6GcM (2/3)**
>
> * **Weaknesses**:
>
> 4. > Can the authors evaluate on MMLU as well?
>
>     To address this, we have conducted additional evaluations on the MMLU benchmark.
>     The results are included in the main table in the experiment sections of the revised paper.
>
> 5. > I would have also liked to see the reduction in FLOPS compared to the standard baseline
>
>     We have included the reduction in FLOPs compared to the baseline in Tab. 1 and Tab. 3 in the revised paper.
>
> 6. > how their scheme is impacted by fine-grained experts. My intuition is that it would be less effective since their is more capacity within the base MoE to learn different types of experts and thus the need for negated or null experts might be less.
>
>     We have included additional experiments using a fine-grained MoE model with 4 activated experts out of a total of 16, where each expert is half the width compared to the base model.
>     We show the experimental results below:
>
>     | Method | Avg. K | \#FLOPs $\downarrow$  | ARC-Easy | BoolQ | MMLU | LAMBADA | HellaSwag | OpenBookQA | PIQA | SIQA | WinoGrande | Avg
>     |:-------|:-|:-|:-|:-|:-|:-|:-|:-|:-|:-|:-|:-|
>     | Top-K baseline | 4.00 | -  | 56.69 | 55.35 | 25.16 | 50.16 | 42.72 | 39.6 | 68.93 | 44.11 | 52.49 | 48.36
>     | TC-MoE | 3.87 | 2.3\% | 57.58 | 58.56 | 26.80 | 50.46 | 43.16 | 41.80 | 68.28 | 45.19 | 52.09 | 49.32
>
>     We observe that our method outperforms the baseline by 0.96\% while reducing FLOPs by 2.3\%.
>
>     Additionally, here is the activated ratio of each type of expert:
>
>     | $K$ | \#Experts | Activated ratio of $E^1$ | Activated ratio of $E^0$ | Activated ratio of $E^{-1}$ |
>     |:-------|:---------:|-------:|-------:|-------:|
>     | 2 | 8 | 86.1% | 4.9% | 9.0% |
>     | 4 | 16 | 91.3% | 5.5% | 3.2% |
>
>     From these results,
>     we can see that the activated ratio of $E^0$ decreases in the fine-grained setting, while the activated ratio of $E^{-1}$ slightly increases.
>     Our intuition is that, with fine-grained experts, negated experts also become more fine-grained and can contribute more effectively. While null experts become less important, as other experts gain more capacity to capture different types of knowledge.
>
>     We have included these results and a detailed explanation in the revised version.
>
>
> 7. > Typos and grammar errors Line 105 - of classical routing scheme
>
>     Thank you for pointing this out.
>     We have corrected the typos and grammar errors in the revised version.

---

> ### Author Response · Authors · 2024-11-21
> **Response to Reviewer 6GcM (3/3)**
>
> * **Questions**:
>
> 1. > Why did the authors not release their code? How can I verify their results in this paper?
>
>     We understand the importance of code availability for verification and reproducibility.
>     We promise to release our code upon publication.
>     To ensure reproducibility, we provide the following details in the paper:
>     * Pre-trained dataset: The dataset used for training is described in Section 4.1.
>     * Model architectures and configurations: Detailed information on the architecture and configurations can be found in Sections 4.1 and 4.2.
>     * Training hyperparameters: We outline all relevant training hyperparameters in Appendix A.2.
>
>     We believe these details will allow others to verify our results and build upon our work.
>
> 2. > In Fig 2B I can see that there is no null expert for Expert 3. This is different from what the text says that the total expert count = 3 x base_num experts. Is this just an illustration error?
>
>     > How is the computation for the total number of experts done? It says on Line 210 it is 2N+K but if each expert is expanded to three experts, isn't that simply 3N? Please clarify this.
>
>     This is not an illustration error.
>     As clarified in lines 206-208 in the original paper (lines 210-213 in the revised paper),
>     since $E_i^0$ has no parameters and is the same across all experts,
>     we simplify by maintaining only $K$ of them,
>     This is sufficient for the Top-K router to activate any number from 0 to $K$ of these experts.
>     Therefore, in Fig. 2b, we have $2$ null experts instead of $3$.
>     The total number of experts is thus $2N + K$ rather than $3N$.
>
> 3. > Is this paper suggesting that a dense forward pass is worse than a sparse forward pass? ... It would be beneficial to be able to add some experiments where the authors can show activation how the performance on a benchmark task(s) is impacted by increasing top-k by 1 over some useful range.
>
>     We indeed suggest that a dense forward pass may be worse than a sparse forward pass in some cases,
>     since we point out that some activations can contribute negatively to the model outputs.
>
>     We also agree with you that increasing the number of activated experts generally results in better performance,
>     as we have observed a similar phenomenon.
>     However, it is important to clarify that this does not contradict our findings.
>     The reason is that the negative activations are not distributed uniformly across tokens.
>     Some tokens only have positive activations, while others have both positive and negative activations.
>     Reducing the number of activated experts impacts every token,
>     and the model's performance may degrade if deactivating positive activations has a larger effect than deactivating negative ones.
>     In conclusion, our paper does not advocate for sparse forward passes for all tokens but specifically for those with negative activations.
>
>     We have included a new figure (Fig. 3b in the revised paper) to demonstrate how performance on HellaSwag is impacted by varying the average number of activated experts.
>     As shown in this figure,
>     performance on HellaSwag improves monotonically as the number of activated experts increases.
>
>
> 4. > How are the results in Fig 1a computed? Are these results on benchmark scores evaluated for a particular setting of K? A more detailed caption would be helpful for future readers.
>
>     Fig. 1a shows the distribution of contributions from each category of activations.
>     The contribution of each activation in this figure is computed by measuring the difference in the quality of model outputs when masking this activation.
>     In Fig. 1a, we use data from the pre-train dataset and the language modeling loss to measure the quality of model outputs.
>     These results are computed on the Top-K baseline model we pre-trained on RedPajama, where $K=2$ and the number of experts is $8$.
>     We have included more detailed captions in the revised paper.
>     More details can be found in Appendix A.1.
>
>     Additionally, we have included figures showing the distribution of contributions from activated experts based on data from the test set (Fig. 5 in the revised paper).
>
>
> 5. > Do the authors find any connection with null experts to attention sinks [1]? ...
>
>     Thank you for this insight.
>     We have addressed this point in conjunction with Weakness 2.
>     For a detailed explanation, please refer to our response to Weakness 2.

---

> ### Comment · Reviewer_6GcM · 2024-11-21
>
> I thank the authors for addressing my concerns, running additional experiments and adding additional clarifications. I have edited my score for the paper to a 6.

---

### Official Review · Reviewer_3e6U · 2024-11-02

**Soundness:** 3
**Presentation:** 3
**Contribution:** 3
**Rating:** 6
**Confidence:** 4

**Summary:**

This paper identifies the limitation of the commonly-used TopK routing scheme of MoEs:  unnecessary activations and underutilization of existing experts. The authors propose a solution by expanding expert space via a ternary set. Experimental results showcase the effectiveness of the method.

**Strengths:**

- The paper is easy to follow and clearly structured.

- The authors propose a novel method for improving MoE, which does not act on the router but takes a different approach to expand expert space.

- The experiments are extensive and the ablation study effectively shows the necessity of having both zero experts and negative experts.

**Weaknesses:**

- Only the LLaMA architecture is tested, and only Top2 routing is performed. It is questionable if the results can generalize to other base architectures and TopK (k=1, 4, etc.) routing.

**Questions:**

- I am having some difficulty understanding how the reward loss functions as you intend. Based on equation (12), minimizing the reward loss seems to result in assigning gating values close to zero for the $E_0$ experts?

---

> ### Author Response · Authors · 2024-11-21
> **Response to Reviewer 3e6U**
>
> * **Weaknesses**:
> 1. > Only the LLaMA architecture is tested, and only Top2 routing is performed. It is questionable if the results can generalize to other base architectures and TopK (k=1, 4, etc.) routing.
>
>     We understand the significance of demonstrating the generalizability of our method across different architectures and routing configurations.
>     Due to computational resource constraints and the significant time required to pre-train multiple architectures,
>     we focus our experiments on the LLaMA architecture for the language modeling task.
>
>     To address concerns about the limitation of testing only Top-2 routing,
>     we conduct additional experiments using a Top-4 routing configuration,
>     where 4 experts are activated out of 16 total experts.
>     The results are summarized below:
>
>     | Method | Avg. K | \#FLOPs $\downarrow$ | ARC-Easy | BoolQ | MMLU | LAMBADA | HellaSwag | OpenBookQA | PIQA | SIQA | WinoGrande | Avg
>     |:-------|:-|:-|:-|:-|:-|:-|:-|:-|:-|:-|:-|:-|
>     | Top-K baseline | 4.00 | - | 56.69 | 55.35 | 25.16 | 50.16 | 42.72 | 39.6 | 68.93 | 44.11 | 52.49 | 48.36
>     | TC-MoE | 3.87 | 2.3\% | 57.58 | 58.56 | 26.80 | 50.46 | 43.16 | 41.80 | 68.28 | 45.19 | 52.09 | 49.32
>
>     The improvements observed in the Top-4 routing configuration are consistent with those observed in the Top-2 routing setting.
>
>     We have included these results in Tab. 1 in the revised paper.
>
>
> * **Questions**:
> 1. > I am having some difficulty understanding how the reward loss functions as you intend. Based on equation (12), minimizing the reward loss seems to result in assigning gating values close to zero for the E0 experts?
>
>     Thank you for pointing this out.
>     Your observation is correct.
>     To address this, we have revised the equation by incorporating a minus sign.
>     The corrected formulation is as follows:
>     $$
>     L_{rwd} = -\frac{1}{T}\sum_{i=1}^{K}\sum_{j=1}^{T} g_{E_i^0}(h_j).
>     $$
>     With this revision, minimizing the reward loss now results in encouraging the router to assign larger gate values for $E^0$ experts.

---

> > ### Comment · Reviewer_3e6U · 2024-11-24
> >
> > Thanks for the response. I would like to keep my original evaluation.

---

### Official Review · Reviewer_7syP · 2024-11-07

**Soundness:** 3
**Presentation:** 4
**Contribution:** 3
**Rating:** 8
**Confidence:** 4

**Summary:**

The authors propose an alternative routing scheme to the widely-used Top-K approach in Mixture of Experts (MoE) models. Their method expands the expert space by introducing a multiplicative set {−1,0,1}, allowing each expert E to activate as E\*1, E\*(−1), or E\*0. Here, E\*0 corresponds to a no-operation choice, effectively reducing the number of active experts to save computational resources. This addition permits the model to activate fewer than K experts when beneficial, lowering computation costs.
To optimize the routing, they adapt the load balancing loss to treat each expert Ei equivalently, regardless of whether it is activated with +1 or −1. Additionally, they introduce a loss term that rewards the model for selecting E\*0, controlled by a hyperparameter that balances efficiency and accuracy. This approach allows the model to expand the effective expert space without adding significant parameter costs. Their results show improvement over three other competitors (Top-K, Random drop, Top-P) in both accuracy and computational savings.

**Strengths:**

- The authors introduce a novel routing scheme that minimizes overhead while achieving comparable or improved accuracy and reduced computational cost.
- The paper is well-written, with a clear, logical flow and excellent presentation.
- The evaluation is thorough, tested across multiple datasets, strengthening the results.
- Additional analyses, such as the impact of unnecessary activations, activated ratio of different expert types, Load Balance, and the ablation study, are relevant and well-executed, adding depth to the study.
- The authors made a commendable effort in providing detailed information for reproducibility, including specific model configurations, dataset details, and experimental setups.

**Weaknesses:**

- Some figures are not fully self-explanatory; they lack clear takeaways, requiring readers to refer back to the text for important details to understand them.
- The improvements in accuracy appear marginal and may not be statistically significant (although it does seem to be better in terms of computation needed).
- Despite reporting the accuracy correctly on some test benchmarks, some of the results presented in the figures (Figs. 3, 4, 5, 6 and 7) are based on training loss or analyses conducted solely on the training dataset. For a more robust evaluation, results should ideally be shown on a validation or test set to confirm generalization.

**Questions:**

- With a variety of datasets used, why were some of the most standard benchmarks, such as MMLU or SuperGLUE, left out?
- Could the authors provide more justification for choosing (K = 2) in their Top-K approach? It appears they reference Zoph et al. (2022), but additional context would clarify this choice.
- Some benchmarks categorize tasks by difficulty. Would it be possible to show if the model's accuracy improvements correlate with task difficulty?
- Why did the authors limit quantization levels to ({-1, 0, 1})? Exploring additional quantization levels might provide further insights.
- Given the new term introduced in the loss function, is the loss comparison with other baselines still fair and directly comparable?

---

> ### Author Response · Authors · 2024-11-21
> **Response to Reviewer 7syP (1/2)**
>
> * **Weaknesses**:
> 1. > Some figures are not fully self-explanatory; they lack clear takeaways, requiring readers to refer back to the text for important details to understand them.
>
>     To address this concern,
>     we have included reader-friendly explanations and clear takeaways in Figs 1, 3, 5, 6, and 7 in the revised version of our paper.
>
> 2. > The improvements in accuracy appear marginal and may not be statistically significant (although it does seem to be better in terms of computation needed).
>
>     We conduct additional experiments using a high-quality dataset, FineWeb.
>     These experiments reveal more significant improvements:
>     When training a tiny model, TC-MoE achieves an average improvement in accuracy by 1.18\% and reduces FLOPs by 5.8\% simultaneously.
>     When training a base model, TC-MoE achieves an average improvement in accuracy of 1.17\% and reduces FLOPs by 5.1\% simultaneously.
>     These results consistently demonstrate that our method is not only effective but also computationally efficient.
>     We have included these new experimental results in Tab. 1 of the revised paper.
>
> 3. > Despite reporting the accuracy correctly on some test benchmarks, some of the results presented in the figures (Figs. 3, 4, 5, 6 and 7) are based on training loss or analyses conducted solely on the training dataset. For a more robust evaluation, results should ideally be shown on a validation or test set to confirm generalization.
>
>     This is indeed a valuable suggestion.
>     For Fig. 3, we have added an additional figure (Fig. 3b in the revised paper) based on the performance on the evaluation dataset.
>     For Figs. 4, 5 and 6, we have included additional figures (Fig. 5, 6b and 7b in the revised paper) based on results from the evaluation dataset (ARC-Easy).
>     For Fig. 7, we include this figure to demonstrate the changing curves of activated ratios during training.
>     The activated ratio during inference can be found in Fig. 6 in the revised version.
>     We have included these additional figures and analyses in the revised paper, which should provide a more robust confirmation of generalization.

---

> ### Author Response · Authors · 2024-11-21
> **Response to Reviewer 7syP (2/2)**
>
> * **Questions**:
> 1. > With a variety of datasets used, why were some of the most standard benchmarks, such as MMLU or SuperGLUE, left out?
>
>     In response to your suggestion, we have conducted additional evaluations on the MMLU benchmark and the BoolQ task within the SuperGLUE benchmark (to be consistent with recent papers [1-3]).
>     The results are consistent with our findings from other evaluation benchmarks.
>     We have included these results in Tab. 1 and Tab. 3 and a detailed explanation in corresponding sections.
>
> 2. > Could the authors provide more justification for choosing (K = 2) in their Top-K approach? It appears they reference Zoph et al. (2022), but additional context would clarify this choice.
>
>     We choose (K=2) in the Top-K baseline because top-2 routing is the most common configuration for modern MoE architectures [4-7].
>     We have added this justification in the revised version of our paper.
>
> 3. > Some benchmarks categorize tasks by difficulty. Would it be possible to show if the model's accuracy improvements correlate with task difficulty?
>
>     To investigate this possibility, we perform experiments on the ARC and MMLU benchmarks, which provide categorizations based on task difficulty.
>     For the ARC benchmark,
>     tasks are divided into an easy set (ARC-Easy) and a challenge set (ARC-Challenge) based on difficulty.
>     For the MMLU benchmark,
>     we select three subjects: Biology, Chemistry, and Mathematics.
>     Each subject is further divided into two difficulty levels: high school and college.
>     Below are the results from our experiments:
>
>     | Benchmark | ARC | ARC | MMLU Biology | MMLU Biology | MMLU Chemistry | MMLU Chemistry | MMLU Mathematics | MMLU Mathematics |
>     |:-------|:-|:-|:-|:-|:-|:-|:-|:-|
>     | Difficulty | Easy | Challenge | High school | College | High school | College | High school | College |
>     | Baseline | 60.19 | 28.92 | 31.0 | 27.8 | 29.1 | 22.0 | 24.3 | 29.0 |
>     | TC-MoE | 60.65 (+0.46) | 29.10 (+0.18) | 31.6 (+0.6) | 31.9 (+4.1) | 28.8 (-0.3) | 20.0 (-2.0) | 25.4 (+1.1) | 29.0 (+0.0) |
>
>     Our method achieves improvements on most of the tasks.
>     However, we do not observe a clear correlation between the accuracy improvements and task difficulty.
>
> 4. > Why did the authors limit quantization levels to ({-1, 0, 1})? Exploring additional quantization levels might provide further insights.
>
>     We chose to limit the quantization levels to {$-1, 0, 1$} because this set is both representative and intuitive. These levels allow us to effectively model positive, zero, and negative contributions, which align naturally with the core idea of our method.
>
>     Moreover, we believe that the router plays a critical role in determining the final quantization level since the output of each expert is finally multiplied by the gate values assigned by the router.
>     Nevertheless, we agree that exploring additional quantization levels may further improve performance, as the router can only assign a gate value ranging from $0$ to $1$.
>     We plan to investigate this direction in future work and appreciate the reviewer’s suggestion.
>
> 5. > Given the new term introduced in the loss function, is the loss comparison with other baselines still fair and directly comparable?
>
>     We apologize for the confusion.
>     In our experiments, the comparisons are based on the language modeling loss, which does not include the new term introduced in our loss function.
>     To clarify, we have revised the figure caption in the updated version.
>
> References:
> [1] - Xue, Fuzhao, et al. "Openmoe: An early effort on open mixture-of-experts language models." arXiv preprint arXiv:2402.01739 (2024).
> [2] - Wang, An, et al. "Hmoe: Heterogeneous mixture of experts for language modeling." arXiv preprint arXiv:2408.10681 (2024).
> [3] - Liu, Liyuan, et al. "GRIN: GRadient-INformed MoE." arXiv preprint arXiv:2409.12136 (2024).
> [4] - Zoph, Barret, et al. "St-moe: Designing stable and transferable sparse expert models." arXiv preprint arXiv:2202.08906 (2022).
> [5] - Jiang, Albert Q., et al. "Mixtral of experts." arXiv preprint arXiv:2401.04088 (2024).
> [6] - Wu, Shaohua, et al. "Yuan 2.0-M32: Mixture of Experts with Attention Router." arXiv preprint arXiv:2405.17976 (2024).
> [7] - Wei, Tianwen, et al. "Skywork-MoE: A Deep Dive into Training Techniques for Mixture-of-Experts Language Models." arXiv preprint arXiv:2406.06563 (2024).

---

> ### Comment · Reviewer_7syP · 2024-11-24
>
> Thank you for addressing my concerns in your response. I appreciate the effort you put into clarifying the points I raised. Congratulations on the extensive and well-executed work presented in this submission.

---

### Official Review · Reviewer_h4gE · 2024-11-12

**Soundness:** 3
**Presentation:** 3
**Contribution:** 3
**Rating:** 6
**Confidence:** 3

**Summary:**

This paper presents an interesting study about Mixture of Experts (MoE). Instead of selecting the top-K experts in the architecture of MoE to be active in prediction, the paper extends the weights of experts to be projected to {−1, 0, 1}, which allows negative contributions from certain experts. To incorporate this idea, it is necessary to enable the router to learn more complex routing strategies during the training process. New loss functions are thus designed. Evaluation was conducted on a decoder-only transformer model, primarily based on the LLaMA architecture. The model consists of 32 transformer layers, each including both an attention layer and an MoE layer. And each MoE layer consists of 8 FFN experts. The evaluation tasks are mainly text-based, such as language understanding and commonsense reasoning. The results demonstrate that the proposed model outperforms mostly the other baselines.

**Strengths:**

S1: The paper is easy to follow. The motivation is clear. The overall design is reasonable.
S2: The evaluation is comprehensive, from the overall performance comparison with baselines on downstream tasks, ablation study to analysis of active experts over layers, over steps and so on.

**Weaknesses:**

W1: In the proposed architecture, will the requirement on memory doubled, or even more? Although no extra computational cost may be introduced.
W2: The results show that most of the time, only 2 experts (or on average 1.8 experts) are selected for being active. Is this small number of active experts limiting the performance improvement regarding the introduction of negative contribution? By checking Table 1 and 2, we can see that the proposed model is not always outperforming baselines with a notable improvement.

**Questions:**

Please see the W1 and W2.

---- After the rebuttal
I have read the response and additional results from authors. W1 and W2 are addressed.

---

> ### Author Response · Authors · 2024-11-21
> **Response to Reviewer h4gE**
>
> * **Weaknesses**:
> 1. > In the proposed architecture, will the requirement on memory doubled, or even more? Although no extra computational cost may be introduced.
>
>     We would like to clarify that the memory requirements are not doubled in our design.
>     Our approach involves augmenting the original expert space by utilizing the ternary set {$-1, 0, 1$},
>     which projects each expert $E_i$ into three distinct experts: {$E_i^{-1}, E_i^0, E_i^1$}.
>     This is defined as follows:
>     $$
>     E_i^{1}(\mathbf{h}) \coloneqq E_i(\mathbf{h}),\quad E_i^{0}(\mathbf{h}) \coloneqq 0,\quad E_i^{-1}(\mathbf{h}) \coloneqq -E_i(\mathbf{h}),\quad \forall \mathbf{h}\in \mathbb{R}^d.
>     $$
>     Importantly, experts $E_i^{1}$ and $E_i^{-1}$ share the same parameters, and the relationship $E_i^{-1}(\mathbf{h}) = -E_i^{1}(\mathbf{h})$ always holds.
>     In practice:
>     * **Shared Parameters**: There is no need to store a separate copy of parameters for $E_i^{-1}$. We store only the parameters of $E_i^{1}$, and when $E_i^{-1}$ is activated, we compute $E_i^{1}(\mathbf{h})$ first and then multiply the output by $-1$ to obtain $E_i^{-1}(\mathbf{h})$.
>     * **No Additional Parameters for $E_i^0$**: Since $E_i^0$ always outputs 0, it does not require any parameters.
>
>     Thus, the memory requirements for storing experts in our method remain the same as those in the baseline model. This efficient parameter sharing ensures that our proposed method achieves its goals without introducing additional memory overhead.
>
> 2. > The results show that most of the time, only 2 experts (or on average 1.8 experts) are selected for being active. Is this small number of active experts limiting the performance improvement regarding the introduction of negative contribution? By checking Table 1 and 2, we can see that the proposed model is not always outperforming baselines with a notable improvement.
>
>     Thank you for pointing out this important aspect.
>     To address your concern,
>     we have included an additional experiment on a fine-grained MoE model with 4 activated experts out of 16 total experts.
>     We show the experimental results as below:
>
>     | Method | Avg. K | \#FLOPs $\downarrow$ | Activated ratio of $E^{-1}$ | ARC-Easy | BoolQ | MMLU | LAMBADA | HellaSwag | OpenBookQA | PIQA | SIQA | WinoGrande | Avg
>     |:-------|:-|:-|:-|:-|:-|:-|:-|:-|:-|:-|:-|:-|:-|
>     | Top-K baseline | 4.00 | - | - | 56.69 | 55.35 | 25.16 | 50.16 | 42.72 | 39.6 | 68.93 | 44.11 | 52.49 | 48.36
>     | TC-MoE | 3.87 | 2.3\% | 5.5\% | 57.58 | 58.56 | 26.80 | 50.46 | 43.16 | 41.80 | 68.28 | 45.19 | 52.09 | 49.32
>
>     Compared to the results with only 2 activated experts,
>     we observe that the average improvement made by TC-MoE increases from 0.83% to 0.96%,
>     and the activated ratio of negative experts increases from 4.9\% to 5.5\%.
>     These results support your hypothesis that increasing the number of active experts amplifies the performance gains brought by the introduction of negative contributions.
>
>     We have included these results in Tab. 1 in the revised paper.

---

> > ### Comment · Reviewer_h4gE · 2024-11-26
> > **Reply to the rebuttal**
> >
> > Thanks for the response and additional experiments. My concerns are addressed. I could   increase my rating to 6.

---

> ### Author Response · Authors · 2024-11-25
> **Could you please check our response?**
>
> Dear Reviewer h4gE,
>
> We would like to express our sincere gratitude for the time and effort you have dedicated to reviewing our manuscript and providing thoughtful feedback. We understand that this is a particularly busy period, and we greatly appreciate the demands on your time.
>
> We have made every effort to carefully address your comments. As the Author-Reviewer discussion period is drawing to a close, we would be truly grateful if you could kindly let us know whether our responses have adequately addressed your concerns.
>
> Once again, thank you for your invaluable contributions to our work.
>
> Best regards,
>
> Authors

---

### Author Response · Authors · 2024-11-21
**Global Response**

We sincerely thank all reviewers for their time and effort in reviewing our paper and providing thoughtful and constructive feedback.

In response to your valuable suggestions, we have made several key updates to enhance the quality and clarity of our work. The major revisions are as follows:
* **Expanded Experiments**: We have included additional experiments on models with a broader range of settings, as presented in Tab. 1, along with detailed discussions in Section 4.2.
* **New Results and Analyses**: We added figures showcasing results on the test set and provided corresponding analyses in Section 4.4 to offer a more comprehensive evaluation of our approach.
* **Improved Captions**: We refined the figure captions to provide clearer and more detailed takeaways.
* **Additional Intuitive Justification**: We have included an intuitive justification for improving the Top-K routing scheme, detailed in Appendix A.3.

All modifications have been highlighted in blue for your convenience.

We sincerely hope these revisions address your concerns and enhance the readability and comprehensiveness of our paper.

---

### Meta-Review · Area_Chair_AVvC · 2024-12-19

**Metareview:**

The proposed approach extends MoE to include negative experts, which in turn yields more complex routing strategies. Experiments demonstrate the superiority over existing MoE baselines over several datasets.

Reviewers found the paper to be novel, particularly w.r.t. to the resulting routing scheme, well-written, while the evaluation was praised for its thoroughness and clarity w.r.t. reproducibility.  Several weaknesses raised were addressed during the rebuttal phase with clarifications and additional experiments, that should certainly be included in the paper. Weaknesses that somewhat remain include the statistical significance in earlier experiments (though new ones appear stronger), and the restriction of experiments to LLaMA, though resource limitations also explain this. The authors are also expected to release their code post-acceptance (they stated they will do so).

**Additional Comments On Reviewer Discussion:**

Reviewers appreciated the expanded experiments on models with a broader range of settings, including increasing the expert number, the explanation regarding memory requirements, the experiments on new datasets,  the scrutiny of non-accuracy metrics also on the test-set, the evaluation w.r.t. flops, the inclusion of the MMLU benchmark, as well as the various additional illuminating discussions/intuitions provided. Overall, the paper has improved significantly post-rebuttal.

---

### Decision · Program_Chairs · 2025-01-22

Accept (Poster)